# Novel Esomeprazole Magnesium-Loaded Dual-Release Mini-Tablet Polycap: Formulation, Optimization, Characterization, and In Vivo Evaluation in Beagle Dogs

**DOI:** 10.3390/pharmaceutics14071411

**Published:** 2022-07-05

**Authors:** Taek Kwan Kwon, Ji-Hyun Kang, Sang-Beom Na, Jae Ho Kim, Yong-Il Kim, Dong-Wook Kim, Chun-Woong Park

**Affiliations:** 1College of Pharmacy, Chungbuk National University, Cheongju 28160, Korea; taekkwan@hanmi.co.kr (T.K.K.); jhkanga@naver.com (J.-H.K.); sbna42@naver.com (S.-B.N.); 2Pharmaceutical Research Centre, Hanmi Pharm. Co., Ltd., Hwaseong 18469, Korea; dbfllove@hanmi.co.kr (J.H.K.); cjkyi@hanmi.co.kr (Y.-I.K.); 3College of Pharmacy, Wonkwang University, Iksan 54538, Korea; pharmengin@gmail.com

**Keywords:** esomeprazole magnesium, solid dosage form, polymeric enteric coating, dual release, Eudragit

## Abstract

Esomeprazole magnesium (EMP) is a proton pump inhibitor (PPI) that reduces acid secretion. EMP has a short plasma half-life (approximately 1.3 h); hence, nocturnal acid breakthrough (NAB) frequently occurs, disturbing the patient’s nighttime comfort and sleep. We aimed to develop a novel esomeprazole magnesium-loaded dual-release mini-tablet polycap (DR polycap) with a prolonged onset time and improved bioavailability to prevent NAB. The formulation of the EPM mini-tablet core resulted in rapid drug release. The core was coated with an inner coating and an Eudragit^®^ L30D-55 aqueous dispersion coating to prepare the first-release mini-tablet. In addition, the core was coated with an inner coating and an aqueous dispersion of Eudragit^®^ S100 and Eudragit^®^ L100 coating to prepare the second-release mini-tablet. Each mini-tablet type was characterized using an in vitro dissolution test and microscopic examination. After testing, 10 of each mini-tablets were placed together in hard capsules to form DR polycaps. The combination of mini-tablets was optimized via in vitro release testing and in vivo pharmacokinetic studies. The AUC_0–24h_ of the DR polycap was similar to that of a comparable commercial product (Nexium^®^); C_max_ was lower by approximately 50%, and T_max_ was extended by approximately 1.7-fold. In conclusion, DR polycap is an alternative to commercial products with improved NAB and dosing compliance because of its dual-release characteristics.

## 1. Introduction

Esomeprazole, a proton pump inhibitor (PPI), reduces acid secretion by inhibiting H^+^/K^+^ ATPase activity in gastric parietal cells. It is widely used alone or in combination with other drugs, such as non-steroidal anti-inflammatory drugs (NSAIDs, e.g., naproxen), for treating gastric ulcers, gastroesophageal reflux disease (GERD), Zollinger–Ellison syndrome, and erosive esophagitis [1], and for the long-term management of patients with peptic ulcers and *Helicobacter pylori* infections [2,3]. Esomeprazole magnesium (EMP), the salt form of esomeprazole, is more stable than esomeprazole, depending on the pH. It decomposes rapidly in acidic media but remains stable under alkaline conditions [4]. Due to the pH effect, most products containing EMP are provided as delayed-release capsules or tablets consisting of an enteric-coated multi-unit pellet system (MUPS). Nexium^®^ (AstraZeneca, Cambridge, UK) is available in the same dosage form containing 20 or 40 mg of esomeprazole [3].

EMP is absorbed in the small intestine, especially the duodenum, and its absolute bioavailability is 89% [5]. Compared to other PPIs, EMP has a short plasma half-life of approximately 1.3 h. Consequently, nocturnal acid breakthrough (NAB) frequently occurs with EMP, resulting in worse nighttime gastric acidity control than during daytime [6]. A limitation of PPIs is that they often do not control acid secretion over an entire 24-h period with a single daily oral dose. Approximately 30% of GERD patients receiving PPI therapy experience treatment failure [7]. In addition, a return of symptoms in the latter part of the 24-h treatment period often occurs with PPI treatment [8]. Therefore, a PPI that provides a sustained effect over a 24-h interval between doses would improve clinical outcomes. In previous studies that improved acid suppression with twice-daily dosing, we observed that many patients using such a regimen experienced gastric acid secretion during nighttime. Twenty milligrams of esomeprazole administered twice daily is superior to 40 mg administered once a day in controlling the stomach’s 24-h acid secretion and pH [9]. However, twice-daily dosing is not an effective method, leading to lower dosing compliance. For avoiding NAB and improving dosing compliance, it is urgent to develop an EMP-controlled release formulation to maintain the drug plasma concentration and prolong the duration. As in a previous study on dexlansoprazole [10], a design is needed to provide a 1st drug peak in the proximal small intestine (duodenum) and a 2nd peak at more distal regions of the small intestine (jejunum) several hours later.

As mentioned above, owing to their acid degradation properties, most products containing EMP consist of enteric-coated MUPS. MUPS offer advantages such as more predictable gastric transit time and drug absorption, consistent action, and less intra-subject variability, reducing the risk of systemic toxicity originating from dose dumping [11,12]. However, MUPS formulations have quality control disadvantages, such as ensuring the uniformity of content and weight and compressing the coated subunits into tablets with sufficient hardness and low friability without damaging the film coating and residual solvents. This consumes process time, affects manufacturing reproducibility and process yield, making it economically less feasible [13,14,15].

Mini-tablets generally refer to tablets with a diameter <2.0 mm—they are small compared to conventional tablets. Their advantage is being of uniform size, so there is little mass deviation. They are relatively easy to manufacture (can be compressed using conventional tablet press using customizing designed multi-tip punches), can be coated to modify drug release, and can be filled in capsules such as other multiple-unit dosage forms. In addition, mini-tablets have excellent production reproducibility, low risk of dose dumping and bioavailability variation, and high dispersibility in the digestive tract, thereby minimizing the risk of high local drug concentrations [16]. Hence, mini-tablets can be a good alternative to pellets, granules, or other MUPS [12]. Finally, modified release systems that improve drug bioavailability can be successfully used in pediatric, gastroretentive, bioadhesive, and oral disintegrating dosage forms [17,18].

Polymer film coatings are widely used for controlled drug release in pharmaceutical formulations [19]. Although several factors influence controlled drug release through polymer film coating, the dispersion of the dosage form and additives in the coating solution and dispersion can play an important role in the release profile of the coated dosage form. Eudragit^®^ is an aqueous coating polymer dispersion containing poly(meth)acrylates well known in the pharmaceutical industry [20]. Eudragit^®^ L30D-55 (ELD-55) is a representative coating polymer that prevents the decomposition of unstable drugs in the gastrointestinal tract acidic solutions and is mainly applied to formulations that target drug release into the upper gastrointestinal tract [21,22]. Eudragit^®^ L100 (EL-100) is a controlled-release coating polymer used for precise drug targeting in the gastrointestinal tract, allowing drug release from the middle to the upper part of the small intestine. It is soluble at a pH of 6.0 or higher. Eudragit^®^ S100 (ES-100) is a functional delayed-release polymer for colonic delivery and gastrointestinal targeting, characterized by its dissolution at pH above 7.0. Coating polymers are often used together for drug release [23,24].

Beagle dogs are generally known to be suitable for pre-clinical studies of oral dosage form. The reason is that the metabolic activity is the most similar to that of humans among various species of animals. Since the beagle dog is a species with similar metabolic activity to humans, the prediction of pharmacokinetics in humans may be more accurate [25]. In many references, it was found that the bioavailability relation between beagle dogs and humans is very high [26,27]. However, there is a disadvantage in that the difference of intestinal environment between beagle dogs and humans. In particular, the pH of the stomach is different, which can be greatly affected in the case of drugs such as EMP, which is unstable to acid. In order to compensate for these shortcomings, many studies using pentagastrin have been conducted [28,29,30]. Pentagastrin lowers the pH of the dog’s stomach to that of a human within 30 min of intramuscular injection and reduces absorption variability in dogs [31].

In this study, we manufactured an EMP-loaded dual-release mini-tablet polycap (DR-polycap). The core was coated with ELD-55 as 1st release mini-tablet for targeting duodenum or ES-100/EL-100 as 2nd release mini-tablet for targeting jejunum (Figure 1A). The 1st release mini-tablet and 2nd mini-tablet were characterized by an in vitro release test. The DR polycap was prepared by filling hard capsules with an optimized combination of tablets. In vivo pharmacokinetic (PK) studies were performed in beagle dogs under fasting conditions to compare commercial products and DR polycaps (Figure 1B). The study of these DR polycaps is expected to improve compliance and NAB with higher concentration in second half than commercial product, and will be able to compensate for the shortcomings of MUPS such as sufficient hardness and low friability.

## 2. Materials and Methods

### 2.1. Materials

EMP was a gift from Lee Pharma Limited (Hyderabad, India). D-mannitol was provided by Roquette (Lesterem, Singapore). Low-substituted hydroxypropyl cellulose (L-HPC) and hypromellose (HPMC) 2910 were purchased from Shin-Etsu (Tokyo, Japan). Croscarmellose sodium (CMC-Na) was obtained from DuPont Nutrition (Newark, DE, USA). HPC L-type (HPC-L) was supplied by Nippon Soda Co., Ltd. (Tokyo, Japan). Sodium stearyl fumarate (SSF) was purchased from JRS Pharma (Polanco, Spain). Talc was purchased from Merck (Darmstadt, Germany). Triethyl citrate (TEC) was purchased from MORIMURA Bros., INC. (Tokyo, Japan). Glycerol monostearate (GMS) was obtained from Gattefosse (Saint-Priest, France). Polysorbate 80 was purchased from CRODA (Seraya, Singapore). Iron oxide red was obtained from VENATOR (Turin, Italy). ELD-55, ES-100, and EL-100 were obtained from Evonik Industries (Darmstadt, Germany). Hard capsules (HPMC) were obtained from SUHEUNG Co., Ltd. (Cheongju, Korea). A commercial product (Nexium^®^ tablet; 40 mg) was purchased from AstraZeneca Korea Co. (Seoul, Korea). All other reagents were of reagent grade and used without further purification. 

### 2.2. Methods

#### 2.2.1. Preparation and Evaluation of EMP Mini-Tablet Core

The EMP mini-tablets were prepared using the dry granulation method. As shown in Table 1, EMP, microcrystalline cellulose, d-mannitol, low-substituted HPC, dibasic calcium phosphate, croscarmellose sodium, HPC L-type, and sodium stearyl fumarate were well mixed. The powder blend was compacted using a roller compactor (Alexanderwerk Inc., Remscheid, Germany) equipped with a roller with a knurled surface. The process parameters were a roller gap of 2.0 mm and a roller pressure of 15.0 MPa. Selected a feed screw speed of 10: 1 (feed screw speed 25 rpm, speed 2.5 rpm) for the commonly used roller speed ratio. For flake sieving, the impeller speed was maintained at 90 rpm, and the screen size was proceeded to 0.8 mm. Granules were then mixed with sodium stearyl fumarate (2.5%) for 5 min and compressed into 2.0 mm mini-tablets (Figure 1A) using customized multi-tip punches (Figure 2). The tablet was compressed at a pressure such that the hardness of the tablet was about 20 N. The bulk density and tapped density were measured using a tapped density tester JV 1000 (Copley Scientific, Nottingham, UK) using the United States Pharmacopeia (USP) method. Carr’s index and the Hausner ratio were calculated using the following equations:Carr’s index = (V_bulk_ − V_tapped_)/V_bulk_ × 100(1)
Hausner ratio = V_tapped_/V_bulk_(2)

The hardness of mini-tablets was measured by hardness tester EH-01 (Electrolab India Pvt., Ltd., Mumbai, India). The friability test was performed by EF-2 (Electrolab India Pvt., Ltd.) using USP method.

The release profile studies were performed at 25 rpm in phosphate buffer (pH 6.8). The reason is that in order to select a mini-tablet core that rapidly disintegrates and releases immediately, it was carried out under the condition of the lowest shear stress.

#### 2.2.2. Preparation of Coated EMP Mini-Tablet

To improve EMP stability, an inner coating was placed between the core and the controlled-release coating layer. The inner coating aqueous solution was composed of HPMC 2910. Talc was added as an anti-sticking agent. The mini-tablet was coated with a fluid bed coater system (GRE-1, GR Engineering, Seoul, Korea) using a bottom spray (Wurster Spray) under the following conditions. The inner coating solution was composed of 11% HPMC 2910. The obtained mini-tablet was coated with an inner coating solution to gain 5% weight under the following coating conditions: nozzle (1.0 mm) attached to a peristaltic pump, spray rate of 6.0 g/min, atomization pressure of 1.2 bar, fluidization air rate of 80–140 m^3^/h, inlet temperature 53 °C, and product temperature 48 °C.

ELD-55, about 30% TEC (weight to dry polymer, serving as plasticizer), about 5% GMS (weight to dry polymer, serving as anti-sticking agent), and approximately 2.0% polysorbate 80 (weight to dry polymer, serving as surfactant) formed the ELD-55 coating dispersion with a solids content of 20%. Before the film-coating process, the blended aqueous dispersion was sieved through a 100 mesh. The ELD-55 coating dispersion was bottom-sprayed onto the inner coated mini-tablet at several coating weight ratios (approximately 9, 11, 13, and 15%, *w*/*w*) (Table 2). The coating process parameters were nozzle (1.0 mm) attached to a peristaltic pump, the spray rate 9.0 g/min, atomization pressure 1.2 bar, fluidization air rate 80–160 m^3^/h, inlet temperature 38 °C and product temperature 33 °C. The ELD-55 coating mini-tablets as 1st release mini-tablets were dried by further fluidizing at 35 °C for 30 min with a curing step and then transferred out of the fluid bed coater.

The ES-100/EL-100 coating dispersion contained 10% (weight to dry polymer) TEC as the plasticizer, 50% (weight to dry polymer) talc as the anti-sticking agent, and 0.3% (weight to dry polymer) iron oxide red as the colorant. The solid content of the dispersions was 40%. The blended solution was sieved through a 100 mesh before the film coating. The ES-100/EL-100 coating was coated on the inner-coated mini-tablet in the same manner as the above-mentioned ELD-55 coating. Several blend ratios of ES-100/EL-100 (1/1, 2/1, 4/1 and 6/1 ratio) and coating ratios (15, 20, 25, and 30%, *w*/*w*) were evaluated (Table 3). The process parameters were set as follows: the same nozzle as the previous coating was used with a spray rate of 8.0 g/min, atomization pressure of 1.2 bar, fluidization air rate of 80–160 m^3^/h, the inlet temperature of 35 °C, and product temperature of 33 °C. After coating, the ES-100/EL-100 coating mini-tablets as 2nd release mini-tablets were dried by further fluidizing at 33 °C for 30 min with a curing step and then transferred out of the fluid bed coater. All operations were protected from light, as the EMP is light sensitive. The coating weight percentage (W%) was calculated using the following equation:W (%) = (MW_b_ − MW_a_)/MW_a_ × 100%(3)
where MW_a_ is the initial mini-tablet weight before coating and MW_b_ is the mini-tablet weight after coating.

#### 2.2.3. Preparation of EMP Loaded Dual Release Mini-Tablet Polycap (DR Polycap)

For the DR polycap (Table 4), 10 ELD-55 coated mini-tablets, and 10 ES-100/EL-100 mini-tablets (EMP 40 mg) were filled with #2 SL HPMC hard capsule using a capsule-filling machine (GKF2500, Bosch; Göttingen, Germany). The filling parameters were set as follows: air pressure 6 bar, vibrator 2 bar, sensor distance 22 mm, and filling performance 30 °C/min.

#### 2.2.4. Morphology of the Coated EMP Mini-Tablets

The morphology of the coated EMP mini-tablets was visualized using image and energy dispersive spectroscopy (EDS) mapping of the cross-sectional mini-tablets. SEM and SEM-EDS mapping were performed using a scanning electron microscope (SEM, ZEISS-GEMINI LEO 1530, Carl Zeiss Co., Ltd., Oberkochen, Germany). In the obtained image, the thickness of the coating layer was measured using ImageJ software at six positions. The sample was coated with platinum using a Hummer VI sputtering device and placed on carbon tape. EDS mapping was performed using the Si atom of talc in the coating layer and the S atom of EMP in the core.

#### 2.2.5. In Vitro Evaluation of DR Polycaps

In vitro drug release from EMP-loaded mini-tablets was evaluated according to the USP 43 monograph “Esomeprazole Magnesium Delayed-Release Capsules” Dissolution Chapter Test 1 method. The EMP mini-tablets were accurately weighed and filled into #2 SL HPMC capsules. Each capsule was placed into a separate metal basket sinker in a vessel containing 300 mL of 0.1 M HCl. The rotation speed of the paddle was 100 rpm, and the temperature of the medium was maintained at 37.0 ± 0.5 °C. For the acid resistance of coated mini-tablets, the mini-tablets were collected on a filter. The residual drug was determined according to the “Acid resistance stage in USP monograph” section. After 2 h, 700 mL of 0.086 M dibasic sodium phosphate pre-equilibrated to 37 ± 0.5 °C was added to each vessel. Then, at predetermined time points, 10 mL samples were withdrawn. The medium was maintained at a constant volume by refilling it with a fresh buffer solution. The withdrawn samples were subsequently filtered through a 0.45 μm membrane filter, and 5 mL of the filtrate was transferred to a suitable glassware containing 1.0 mL of 0.25 M sodium hydroxide and assayed for the dissolved esomeprazole magnesium concentration by UV/Vis spectrophotometry at 302 nm as described above. The test lasted for 4 h in total. All tests were performed with six capsules, and the mean values are reported.

A high-performance liquid chromatography (HPLC) system (Waters Corp., Milford, CT, USA) was used for the quantitative analysis of esomeprazole magnesium. The column used was Inertsil ODS-3V (150 mm × 4.6 mm, 3 μm, GL Science, Tokyo, Japan). The mobile phase consisted of 350 mL of acetonitrile and 500 mL of pH 7.3 phosphate buffer (40.5 μM) in a 1000 mL volumetric flask, diluted to the volume with distilled water, filtered through 0.2 μm nylon membrane filter (Millipore, Billerica, MA, USA), and further degassed before use. The mobile phase was pumped through the column at a 1.0 mL/min flow rate. The column temperature was set to room temperature, and the detection wavelength was 302 nm. The HPLC method was validated for selectivity, sensitivity (Appendix A), linearity (Appendix A), accuracy, precision (Appendix A), and recovery (Appendix A) following the U.S. Food and Drug Administration (FDA) industry guidelines.

#### 2.2.6. In Vivo PK Study of DR Polycaps

Male beagle dogs (8–9 months old, 9.2 ± 1.2 kg; Marshall Co., Beijing, China) were used in all experiments. The dogs were kept in a temperature-controlled environment with a 12 h light/12 h dark cycle for six days. The animals received CANINE food (Cargill Agri Purina, Inc., Seongnam, Korea) and purified water ad libitum. Animal studies were performed following the National Institutes of Health (NIH) Policy and Animal Welfare Act and with the approval of the Institutional Animal Care and Use Committee (IACUC) at the Hanmi Research Centre (AECQ0092, 2017).

We evaluated and compared the PK profiles of DR-1, DR-2, and commercial products (Nexium^®^ 40 mg). Group 1 (n = 6) was administered Nexium^®^ 40 mg, while groups 2 and 3 (n = 6) were administered once DR-1 and DR-2, respectively. Pentagastrin solution was prepared by dissolving 2.4 mg of pentagastrin in 1 mL of dimethyl sulfoxide (DMSO) and diluted 100-fold with normal saline. Each group was injected with 6 μg/kg pentagastrin 30 min before oral administration of 40 mL of water. The pH of fasted dog stomach ranges from pH 1.5 to 6.7, while the pH of the fasted human stomach is approximately 1.7 [29,30]. Pentagastrin lowers the dog stomach pH within 30 min after intramuscular injection, simulating the pH of the human stomach and reducing the variability in absorption in dogs [31]. In addition, one study suggested that when healthy dogs were administered EMP without pentagastrin, two different populations were observed: those that demonstrated lag absorption and those that did not [32]. After administration, approximately 1.0 mL of blood was collected from the dogs’ head vein at predetermined intervals. Blood samples were centrifuged at 12,000 rpm for 2 min at 4 °C using a centrifuge (5415C; Eppendorf, Hamburg, NY, USA) and stored at −70 °C.

All operations were performed under protection from light. Frozen dog plasma samples were thawed at ambient temperatures. Approximately 30 μL of plasma samples were mixed well with 10 μL of internal standard solution and 2 μg/mL lansoprazole in a methanol/water mixture (50:50 *v*/*v*). Methyl tert-butyl ether (0.5 mL) was added to this mixture, followed by sonication for 10 min, and centrifugated at 12,000 rpm for 10 min. The supernatants were evaporated using nitrogen gas at 40 °C, and the residues were dissolved in 1 mL of an acetonitrile/water mixture (50:50 *v*/*v*). Subsequently, 10 μL of the mixture was added to 300 μL of an acetonitrile/water mixture (50:50 *v*/*v*). The supernatant (5 μL) was injected into an ultra-performance liquid chromatography-tandem mass spectrometry system (Waters Xevo TQ-S UPLC-MS-MS system; Waters) to assess the concentration of EMP in the plasma. An electrospray ionization interface in positive ion mode ([M + H]^+^) was installed in the system. The MS-MS parameters were as follows: capillary voltage, 3.0 kV; source temperature, 150 °C; desolvation temperature, 350 °C; desolvation gas flow, 800 L/h; cone gas flow, 50 L/h. The column was HSS CYANO column (Waters, 2.1 mm × 100 mm, 1.8 μm) at 35 °C. The mobile phase was an acetonitrile/5 mM ammonium acetate mixture (70:30 *v*/*v*) with a flow rate of 0.20 mL/min. Qualification was conducted by multiple reaction monitoring (MRM) of the protonated precursor ions and related product ions via the ratio of the area under the peak for each solution and a weighing factor of 1/X^2^. A calibration curve was established over the range of 1–5000 ng/mL in plasma (R^2^ = 0.9987), with a lower limit of quantitation of 1 ng/mL. The method was demonstrated to be suitable, showing intra- and inter-day differences within an acceptable range and good sample stability.

Non-compartmental analysis (WinNonlin; professional edition, version 2.1; Pharsight Co., Mountain View, CA, USA) was performed to calculate the area under the drug concentration-time curve (AUC_0–24h_) from zero to infinity, time taken to reach the maximum plasma concentration (T_max_), maximum plasma concentration of the drug (C_max_), half-life (t_1/2_), and elimination constant (K_el_) with all points.

### 2.3. Statistical Analysis

Statistically significant differences were evaluated by one-way analysis of variance (ANOVA) with LSD or Games–Howell post hoc test using SPSS version 23 (SPSS Inc., Chicago, IL, USA). Statistical significance was set at *p* < 0.05.

## 3. Results and Discussion

### 3.1. In Vitro Evaluation EMP Mini-Tablet

#### 3.1.1. Flowability Test of EMP Mini-Tablet Core Granule

EMP-loaded mini-tablets were prepared according to the formula in Table 1. Mini-tablets granule flowability is important because granules must be well filled with a small dye [33]. In Table 5, through Carr’s index and Hausner ratio, the granule confirmed sufficient flowability to compress mini-tablets. In addition, the hardness and friability of the mini-tablet was secured enough to have no problem in the polycap manufacturing process. Therefore, considering the release profile and granule flowability, C-4 was selected as the mini-tablet core formula.

#### 3.1.2. Dissolution Test of EMP Mini-Tablet Core

The release profile studies were performed at 25 rpm in phosphate buffer (pH 6.8). Each formula profile is shown in Figure 3. When L-HPC was used as an excipient (C-4), a high dissolution rate was observed. There was no effect on the release behavior by the disintegrant (CMC-Na, C1, and C-2) and diluent (dibasic calcium phosphate, C-3). In all formulas, disintegration was completed within 1 min. However, in all formulas except for C-4, it was confirmed that the release rate of EMP was low due to the coning phenomenon of the insoluble ingredients, microcrystalline cellulose and dibasic calcium phosphate (45 min in 25 rpm, then complete drug release at 200 rpm, data not shown). However, in the case of L-HPC [34], it was confirmed that the release was increased due to the swelling in water. 

#### 3.1.3. Dissolution Test of Coated EMP Mini-Tablet

The results of the release profile according to the ELD-55 coating ratio (1st release coating) are shown in Figure 4A. As illustrated, the release behavior of 1st release mini-tablet depended on the coating ratio, and a rapid dissolution profile of more than 80% in 125 min was confirmed in I-2 (11% coating ratio). We tried to optimize the minimum amount of coating with acid resistance up to 120 min. Therefore, I-2 was selected optimal formulation. On the other hand, in the case of Nexium^®^, a commercial product with an enteric coating tablet, 90% was released in 135 min, showing a slower release profile than 1st release mini-tablet. In the case of 9% coated I-1, the EMP decomposed in the acid resistance evaluation within 120 min owing to the low coating rate, and a low release rate was observed after transferring to pH 6.8 medium. The release rate depends on the polymer’s degree of ionization, and the polymer’s dissolution involves the processes of water absorption, swelling, and entanglement [35,36]. The drug was released as soon as the coating polymer was dissolved, and the thicker the polymer, the longer the dissolution time of the polymer. This phenomenon is also associated with the formation of cracks in polymer films. For film-controlled drug release, physical stability, such as film coating thickness and hydrostatic pressure, determines whether the polymer film cracks [37,38]. When the medium diffuses into the mini-tablet core, it creates monotonically increasing hydrostatic pressure inside the tablet, causing the film to crack early during the dissolution test on the thinnest side of the film coating (e.g., edge regions) [39].

The 2nd mini-tablet release profile according to the coating ratio was investigated (Table 3). The controlled release with lag time was confirmed based on the coating ratio (Figure 4B). II-1 coated at a 15% ratio was acid-resistant for 120 min, and after changing the pH to 6.8 medium, drug release started, releasing about 80% of the drug at 180 min. The release was consistently extended as the coating ratio increased. In addition, drug release was completed within 240 min for all coating ratios. However, in 30% with a high ratio (II-4), the release started after approximately 240 min, the dissolution rate was over 85% at 360 min, and the drug release was completed at 420 min. This effect of the coating ratio on the release profile can also be explained by the above-mentioned cracks in the polymer film. Once cracks form in the weak areas of the coating, drug release is controlled primarily through diffusion through water-filled channels instead of through polymer films [23]. This resulted in minimal lag times for the 2nd release mini-tablet targeted to the jejunum at relatively intact II-3 and II-4. However, in the case of a lag time of more than 180 min, there is a possibility that the residence time in the GI tract is prolonged and the tablet is lost during excretion [40]. In the current study, for absorption in the jejunum, the coating ratio was set to 25% (II-3). The effect of the copolymer ratio of ES-100/EL-100 in the coating formulation on the drug release profile is shown in Figure 4B. The mini-tablets coated in the same polymer ratio (II-5) started drug release after a lag time of 180 min, and drug release was completed within 240 min. The release rate decreased significantly as the proportion of ES-100 in the formula increased. There was a direct relationship between the decrease in dissolution rate and the increase in ES-100 content in the formula. As expected, the tablet coated with the highest ratio of ES-100 (II-7) showed a significantly slower dissolution rate than the other formulations (no drug release within 240 min); a relatively large dissolution deviation was confirmed. In contrast, the combination formula containing approximately 66% ES-100 (II-3) released approximately 90% or more of the drug within the same time point (300 min). In addition, in case of II-6, EMP was released from 180 min, reaching about 90% release at 360 min. Consistent with previous reports on tablets coated with the ES-100/EL-100 combination, the release mechanism could be explained by pore formation [41,42]. The polymer of this combination composed of a pH-dependent mechanism coating base showed that the higher the amount of EL-100 within the studied pH (pH 6.8), the weaker or more pores in the coating film were formed so that the dissolution medium could penetrate the tablet. This creates channels and results in faster dissolution of the drug. The coated tablet with the same coating ratio, II-5, released more than 80% of the drug within 180 min in a pH 6.8 medium. This indicates a rather rapid delivery release profile, especially given the average GI transit time for solid dosage forms of jejunum-targeted drugs [24]. In contrast, II-7 (6/1 coating ratio) showed a delayed-release profile compared to II-3 or II-6, which may be due to drug degradation and excretion at intestinal pH. A wide range of drug release behaviors can be achieved by varying the ratio of ES-100/EL-100 in the formula [43]. The relatively low ES-100 content resulted in insufficient release over time. As mentioned in previous studies, the ratio may vary depending on the type of polymer applied, layer thickness, physicochemical properties of the drug, loading capacity, and the size and shape of the tablet [40]. We tried to develop a formulation that dissolution proceeds between 180 min (duodenum) and 360 min (jejunum). Therefore, in further studies, ES-100/EL-100 ratios of 2/1 and 4/1 formulas (II-3 and II-6) were used.

#### 3.1.4. Morphology Characterization of Coated EMP Mini-Tablet

As shown in the SEM image and SEM-EDS mapping in Figure 5, it appears that the coating of the EMP mini-tablet is relatively uniformly, including the edges and walls. The coated mini-tablets exhibited a nearly square shape with convex edges (Figure 5A,B). The average layer thickness of the I-2 walls was 44.1 ± 2.4 μm, but the edges were 35.1 ± 1.4 μm, confirming that the edges portion were relatively thin. Similarly, in II-3, the average thickness was 83.5 ± 1.1 μm, while the edges were 70.0 ± 4.8 μm). In the SEM-EDS image (Figure 5C), C atoms were confirmed to be present in both the core and coating layer. Si atoms (Figure 5D) that exist only in the coating layer were not found in the EMP core. It appears that the coating agent did not penetrate the inside of the core during the coating process and was only on the surface of the EMP core. The S atoms were present only in the EMP core (Figure 5E). A gap exists between S and Si in the merged images (Figure 5F), presumed to be a sub-coating layer. In conclusion, for the dual release of EMP in this study, the minimum coating film thicknesses of 1st release and 2nd EMP mini-tablets were 35.1 ± 1.4 μm and 70.0 ± 4.8 μm, respectively.

#### 3.1.5. Dissolution Test of DR Polycap

DR polycaps of DR-1 and DR-2 were constructed (Table 4). In this case, 10 tablets each of 1st release mini-tablet (I-2) and 2nd release mini-tablet (II-3 or II-6) were filled into #2 SL HPMC capsules (1:1 ratio, EMP 40 mg), and the release profile under the same conditions was determined (Figure 6). Both DR-1 and DR-2 met the USP criteria for the enteric performance test of EMP in 0.1 N HCl (for 120 min) and demonstrated good acid resistance. The formula showed a lag time of 125 min. Then, 40% or more of EMP was rapidly released. This means that the 1st release mini-tablet was completely released within 15 min. After 180 min, the dissolution of the 2nd release mini-tablet portion started slowly. DR-1 showed a release of approximately 90% of EMP within 240 min, whereas DR-2 showed a relatively lower release of approximately 62%. For a formulation with a high ES-100 ratio, the drug release seems to be relatively slow, as the medium penetration into the inside of the tablet is slowed through the coating layer. In both formulations, the release was complete at 360 min. During follow-up in the intestinal state, 120 min after emptying in the gastric state, drug release was rapid, with approximately 100% release in 15 min (1st release). In addition, a continuous, timed-release was confirmed (2nd release). This dual-release profile confirmed the potential for drug release from the duodenum and jejunum targets.

### 3.2. In Vivo Evaluation of DR Polycaps

The PK was determined after oral administration of the EMP-loaded DR polycap and the commercial product (Nexium^®^) at an equivalent dose of 40 mg EMP to beagle dogs. The plasma concentration versus time profiles obtained after a single dose are shown in Figure 7, and the pharmacokinetic parameters are summarized in Table 6. AUC_0–24h_ is 1547.73 ± 458.54 ng∙h/mL for Nexium^®^, 1484.46 ± 401.92 ng∙h/mL for DR-1, and 1302.10 ± 309.64 ng∙h/mL for DR-2, respectively. At AUC_0–24h_, DR-1 was similar wit Nexium^®^, but DR-2 showed more than 15% lower values. DR-1 and DR-2 showed relatively different AUC_0–24h_ pharmacokinetic differences. Due to the difference in the ES-100 content of the 2nd release mini-tablets of DR-1 and DR-2, the 2nd release of DR-2 showed a lower and slower release profile than DR-1. The inhibitory effects of PPI drugs, such as EMP, on gastric secretion are associated with AUC_0–24h_ [44,45]. C_max_ is 708.26 ± 139.70 ng∙h/mL for Nexium^®^, 380.99 ± 124.91 ng∙h/mL for DR-1, and 306.38 ± 92.63 ng∙h/mL for DR-2, respectively. T_max_ is 2.08 ± 0.29 h for Nexium^®^, 3.71 ± 0.96 h for DR-1, and 3.67 ± 1.39 h for DR-2, respectively. Compared with DR-1 and DR-2, C_max_ is also formed by the 2nd release, so C_max_ of DR-2 is about 20% lower than that of DR-1. Compared with Nexium^®^, the DR polycap has an approximately 1.7-fold prolonged T_max_. The T_1/2_ values were 2.59 ± 0.55, 3.34 ± 0.64, and 3.01 ± 0.24 h, respectively. Due to the T_max_, DR-1 and DR-2 had longer half-lives than Nexium^®^. However, statistical significance was found only in DR-1. In addition, due to the relatively high C_max_, DR-1 had a longer half-life than DR-2, but it was not statistically significant. The K_el_ of the Nexium^®^, DR-1, and DR-2 groups were 0.28 ± 0.06, 0.21 ± 0.04, and 0.24 ± 0.04 h^−1^, respectively. Furthermore, in K_el_, both DR-1 and DR-2 showed significantly lower values than Nexium^®^. As with T_1/2_, there was no significant difference between DR-1 and DR-2. In the case of DR-1, the dual release pattern was confirmed through in vitro dissolution and exhibited a dual peak pharmacokinetic profile in vivo (1st peak: approximately 2 h, 2nd peak: approximately 5 h). The drug is adequately absorbed in the small intestine and jejunum when GI transit times through the gastrointestinal tract are considered. Based on these results, we suggest that DR polycaps are bioequivalent to commercial products when tested in beagle dogs, and are expected to improve medication compliance by providing delayed absorption time. In addition, the long half-life may increase the duration of action of the EMP, which is expected to be effective in inhibiting NAB.

## 4. Conclusions

In this study, we produced relatively easily made mini-tablets to construct a dual-release polycap that can rapidly relieve clinical symptoms and maintain the long-term therapeutic effects of EMP. The release behavior was optimized in vitro using ELD-55 as a coating agent for 1st release mini-tablet and ES-100/EL-100 as coating agents for 2nd release mini-tablet. In vitro characterization revealed that the DR polycap had the desired dual release profile. In vivo experiments confirmed similar AUC_0–24h_ and prolonged T_max_ compared to those of a commercial product. In addition, it is expected that the half-life and the rate of elimination constant will also be improved, resulting in a longer duration of action. Pharmacokinetic results indicated that EMP-loaded DR polycap might be an optimal formulation to treat gastric acid-related conditions and prevent nocturnal acid breakthrough.

## Figures and Tables

**Figure 1 pharmaceutics-14-01411-f001:**
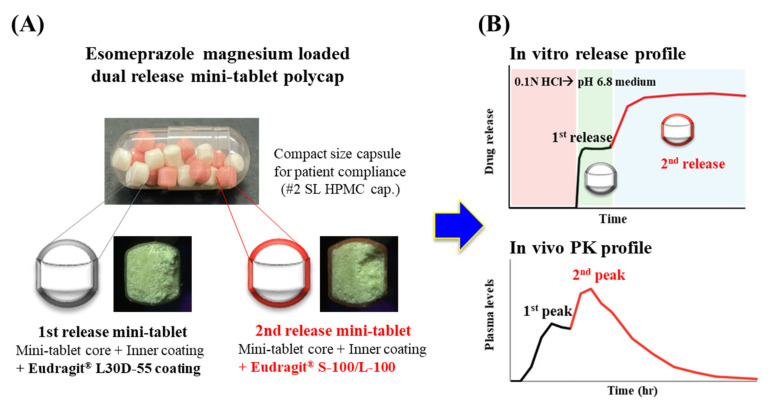
Schematic diagram of esomeprazole magnesium dual release mini-tablet polycap (DR polycaps). (**A**) image of DR polycaps. (**B**) In vitro release and in vivo pharmacokinetics profile of DR polycaps.

**Figure 2 pharmaceutics-14-01411-f002:**
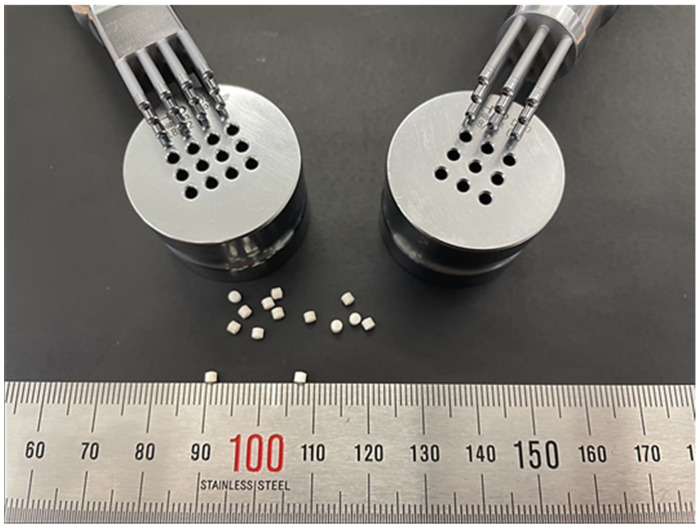
Mini-tablet (2 mm) and multi-tip punches design (left: 12-tip, right: 9-tip).

**Figure 3 pharmaceutics-14-01411-f003:**
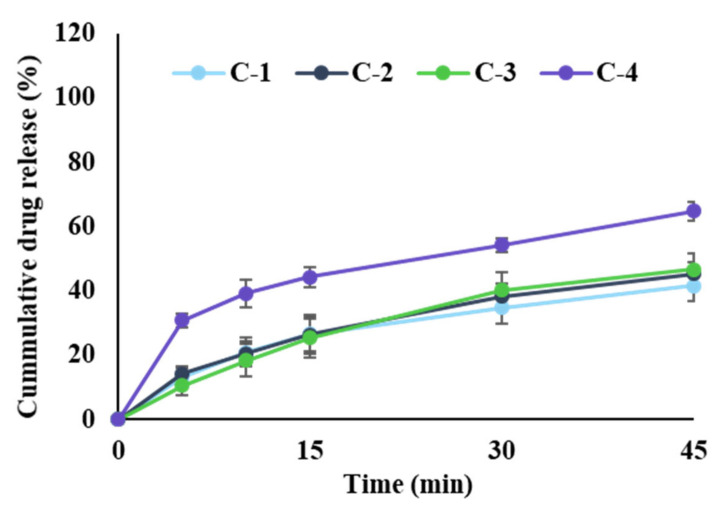
The release profile of EMP mini-tablet core (mean ± standard deviation, n = 6).

**Figure 4 pharmaceutics-14-01411-f004:**
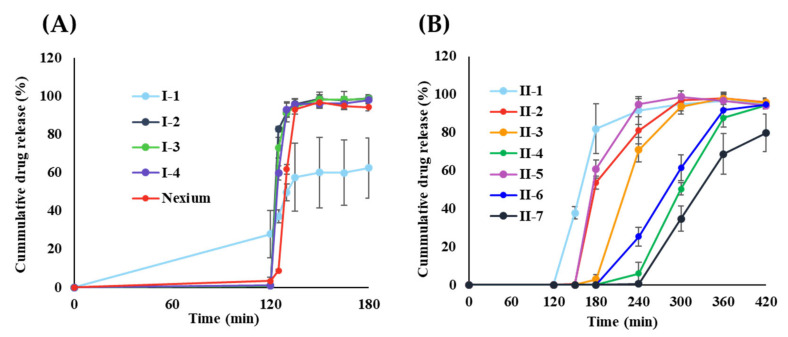
The release profile of coated EMP mini-tablet (mean ± standard deviation, n = 6). (**A**) ELD-55 coated EMP mini-tablet (1st release mini-tablet) with different coating ratio; (**B**) ES-100/EL-100 coated EMP mini-tablet (2nd release mini-tablet) with different coating ratio and different ratio of ES-100/EL-100.

**Figure 5 pharmaceutics-14-01411-f005:**
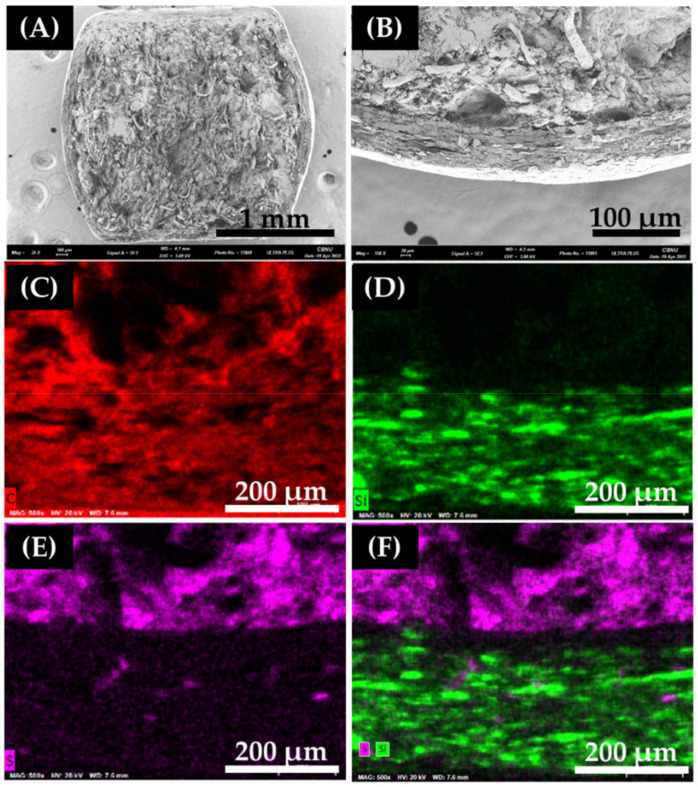
SEM image and SEM-EDS mapping of coated EMP mini-tablet (Formulation code: I-2). (**A**) Cross-section (Magnification: ×36), (**B**) Coating layer (Magnification: ×150). (**C**–**E**) shows SEM-EDS mapping of C, Si, and S atom, respectively (Magnification: ×500). (**F**) Merged image of (**D**,**E**) (Magnification: ×500).

**Figure 6 pharmaceutics-14-01411-f006:**
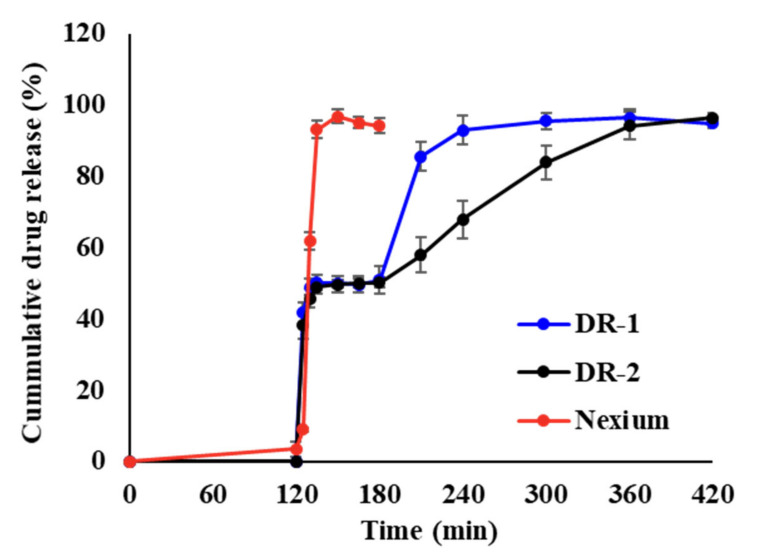
The release profile of DR polycap.

**Figure 7 pharmaceutics-14-01411-f007:**
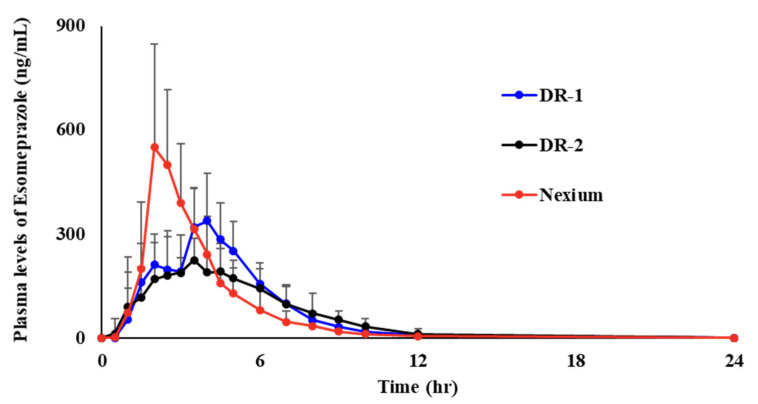
Mean plasma concentration of EMP versus time after a single administration of commercial product (Nexium^®^), DR-1, and DR-2 at the equivalent dose of 40 mg EMP.

**Table 1 pharmaceutics-14-01411-t001:** Composition of EMP-loaded mini-tablet core formulas (Amount per capsule containing 10 mini tablets, unit: mg).

Ingredient	Formulation Code			
	C-1	C-2	C-3	C-4
Esomeprazole magnesium trihydrate	22.3	22.3	22.3	22.3
Microcrystalline cellulose	14.8	13.8	-	-
D-mannitol	30.5	28.7	28.7	28.7
Dibasic calcium phosphate	-	-	13.8	-
Low-substituted hydroxypropyl cellulose	-	-	-	13.8
Croscarmellose sodium	2.4	4.8	4.8	4.8
Hydroxypropyl cellulose (L-type)	2.4	2.4	2.4	2.4
Sodium stearyl fumarate	3.0	3.0	3.0	3.0
Total weight	75.0	75.0	75.0	75.0

**Table 2 pharmaceutics-14-01411-t002:** Composition of EMP-loaded ELD-55-coated mini-tablet as 1st release mini-tablet. (Amount per capsule containing 10 mini-tablets, unit: mg).

Ingredient		Formulation Code			
		I-1	I-2	I-3	I-4
Core	Esomeprazole magnesium trihydrate	22.3	22.3	22.3	22.3
	D-mannitol	28.7	28.7	28.7	28.7
	Low-substituted hydroxypropyl cellulose	13.8	13.8	13.8	13.8
	Croscarmellose sodium	4.8	4.8	4.8	4.8
	Hydroxypropyl cellulose (L-type)	2.4	2.4	2.4	2.4
	Sodium stearyl fumarate	3.0	3.0	3.0	3.0
Inner coating layer	Hypromellose 2910	3.8	3.8	3.8	3.8
	Talc	0.1	0.1	0.1	0.1
	(Water)	(34.0)	(34.0)	(34.0)	(34.0)
ELD-55 coating layer	Methacrylic acid:Ethyl acrylate Copolymer (1:1) 30% dispersion	23.70 (7.11)	28.96 (8.69)	34.23 (10.27)	39.49 (11.85)
	Triethyl citrate	2.13	2.60	3.07	3.54
	Glycerol monostearate	0.38	0.47	0.55	0.64
	Polysorbate 80	0.15	0.19	0.22	0.26
	(Water)	(15.70)	(19.19)	(22.68)	(26.17)
	Total weight	88.67	90.84	93.01	95.18

**Table 3 pharmaceutics-14-01411-t003:** Composition of EMP loaded ES-100/EL-100 coating mini-tablet as 2nd release mini-tablet. (Amount per capsule containing 10 mini-tablets, unit: mg).

Ingredient		Formulation Code						
		II-1	II-2	II-3	II-4	II-5	II-6	II-7
Core	Esomeprazole magnesium trihydrate	22.3	22.3	22.3	22.3	22.3	22.3	22.3
	D-mannitol	28.7	28.7	28.7	28.7	28.7	28.7	28.7
	Low-substituted hydroxypropyl cellulose	13.8	13.8	13.8	13.8	13.8	13.8	13.8
	Croscarmellose sodium	4.8	4.8	4.8	4.8	4.8	4.8	4.8
	Hydroxypropyl cellulose (L-type)	2.4	2.4	2.4	2.4	2.4	2.4	2.4
	Sodium stearyl fumarate	3.0	3.0	3.0	3.0	3.0	3.0	3.0
Inner coating layer	Hypromellose 2910	3.8	3.8	3.8	3.8	3.8	3.8	3.8
	Talc	0.1	0.1	0.1	0.1	0.1	0.1	0.1
	(Water)	(34.0)	(34.0)	(34.0)	(34.0)	(34.0)	(34.0)	(34.0)
ES-100/ EL-100 coating layer	Methacylic acid: Methyl methacrylate copolymer (1:2)	7.87	10.50	13.12	15.75	9.84	15.74	16.87
	Methacylic acid: Methyl methacrylate copolymer (1:1)	3.94	5.25	6.56	7.87	9.84	3.94	2.81
	Talc	5.90	7.87	9.84	11.81	9.84	9.84	9.84
	Triethyl citrate	1.17	1.55	1.94	2.33	1.94	1.94	1.94
	Iron oxide red	0.04	0.05	0.06	0.08	0.06	0.06	0.06
	(EtOH)	(151.72)	(202.29)	(252.87)	(303.44)	(252.87)	(252.87)	(252.87)
	(Water)	(16.62)	(22.17)	(27.71)	(33.25)	(27.71)	(27.71)	(27.71)
	Total weight	97.82	104.12	110.43	116.73	110.43	110.43	110.43

**Table 4 pharmaceutics-14-01411-t004:** Combination of EMP-loaded DR polycaps containing 10 of 1st release mini-tablet and 10 of 2nd release mini-tablet.

Formulation	DR-1	DR-2
Mini-tablet core	C-4	C-4
1st release mini-tablet	I-2	I-2
2nd release mini-tablet	II-3	II-6

**Table 5 pharmaceutics-14-01411-t005:** Granule and tablet characteristics of EMP-loaded mini-tablet core formulas.

Parameter	C-1	C-2	C-3	C-4
Carr’s index	31.15	30.91	32.73	29.63
Hausner ratio	1.31	1.31	1.33	1.30
Hardness (N)	18.6	19.6	21.5	21.5
Friability (%)	0.13	0.13	0.16	0.08

**Table 6 pharmaceutics-14-01411-t006:** The pharmacokinetic parameters of commercial product and DR polycap (DR-1 and DR-2) after single oral administration in beagle dogs (n = 6).

Parameters	Commercial Product	DR-1	DR-2
AUC_0–24h_ (h·ng/mL)	1547.73 ± 458.54	1484.46 ± 401.92	1302.10 ± 309.64
C_max_ (ng/mL)	708.26 ± 139.70	380.99 ± 124.91 **	306.38 ± 92.63 **
T_max_ (h)	2.08 ± 0.29	3.71 ± 0.96 **	3.67 ± 1.39 *
T_1/2_ (h)	2.59 ± 0.55	3.34 ± 0.64 *	3.01 ± 0.68
K_el_ (h^−1^)	0.28 ± 0.06	0.21 ± 0.04 **	0.24 ± 0.04 *

* ANOVA, *p*-value < 0.05 compared with Commercial product; ** ANOVA, *p*-value < 0.005 compared with Commercial product.

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
