# Peer review of "Novel Esomeprazole Magnesium-Loaded Dual-Release Mini-Tablet Polycap: Formulation, Optimization, Characterization, and In Vivo Evaluation in Beagle Dogs"

_pharmaceutics, 2022, doi:10.3390/pharmaceutics14071411_

Round 1

Reviewer 1 Report

Interesting article

Author Response

Thank you for your thorough review and salient observations of this manuscript 

Reviewer 2 Report

The article is interesting and has the potential to be beneficial in advancing the development of dual-release mechanisms for the delivery short half-life drugs. There is however a number of queries that the authors need to address as well numerous spelling and grammar errors in the manuscript. A list of comments can be found below:

1    1. The length of the introduction should be decreased with focus given to the shortcomings of EMP treatment, the current limitation into EMP drug delivery systems and the novelty of the research. The background information regarding the derivatization of EMP and the properties of the polymers used are known and can be removed. Further background with regards to the modified delivery of PPIs should also be provided.

2.     The authors should state in the introduction which areas of the small intestine EMP is absorbed from to correlate it to the dexlansoprazole study that has been cited.

3.     Further information on the novelty of the developed platform and its improvement over previous research should be provided in the introduction.

4.     Figure 1 should be referred to in greater detail in text where the description of the delivery platform has been provided. Figure 1 can be annotated further to reflect the individual components as described in text.

5.     Table 1: the authors should state that C-1, C-2, C-3 and C-4 are the formulation options tested in text. A schematic of the minitablet core would also provide greater detail and understanding with regards to the formulation process.

6.     The author’s have stated the following line in the formulation of the minitablet core: “All formulations of mini-tablets showed approximately 2 kp hardness and less than 0.2 % friability, and an in vitro dissolution test was performed (Table 1)”. The methods for the hardness testing and in vitro dissolution studies need to be included at this point. Additionally, Table 1 does not provide any information with regards to the hardness and drug release studies. The authors should rephrase this sentence.

7.     All tables should be placed after the text in which they have been described.

8.     The morphology method should be separate from the in vitro drug release study methodology.

9.     The in vitro release studies for minitablet core and the coated EMP minitablets have only been undertaken for 45 and 420 minutes respectively. As this system is intended for a prolonged release, the authors should state why the systems were not tested for a time period more applicable to their applications.

10.  A discussion on the hardness of the developed minitablets has not been included in the manuscript, only briefly mentioned in 3.1.1.

11.  The standard deviations for the coated EMP minitablets are large. The authors should discuss this in further detail.

12.  In vivo results. The authors should discuss the therapeutic drug levels of EMP and if the levels reached in vivo would be of a concentration required to exert a therapeutic effect. This is needed to validate that the developed systems are superior to the commercial product. A greater discussion with regards to DR-1 vs. DR-2 should also be provided.

13.  Ethical clearance for this study has not been provided. Additionally, the Informed Consent Statement is not applicable to this study.

14.  A thorough proof-read of the manuscript is suggested as there are numerous spelling and grammar errors.

Author Response

Thank you for your thorough review and salient observations of this manuscript and for the comments and suggestions, which help to improve the quality of this manuscript. Our response follows.

Reviewer 3 Report

The manuscript by Kwon et al. contains a lot of previously unknown data, demonstrating that even traditional solutions are suitable for a fine-tuned controlled release. Even though most of the manuscript is acceptable, there are many minor flaws that only a major revision can entirely correct.

- In the Abstract, the authors defined the half-life abbreviation but later in this section, they did not use that, i.e., writing t½ is useless.

- The Keywords section is a bad joke! Were they able to repeat the words in the title and 'eudragit' so many times? Is this a product of seven authors? Apart from the incorrect Keywords section, the identification of eudragits in the experimental description is different.

- Though the referee accepts that the PK curve in Fig. 1 is strongly idealized and serves demonstration purposes only, a better correlation with Fig. 7 could be more scientific and less misleading.

- Section 2.2.1, Table 1, what does 10T mean? Define it upon the first appearance. If it is an average content for one tablet (calculated from ten tablets), please write it. It is not only unclear that the inserted values are for one tablet only, but are the values analytical (measured) or weighed-in values? On the other hand, in the author's formulation, the esomeprazole content is different (~21 mg) from the standard tablets (20 and 40 mg). This difference is near 5% (for the 20 mg official content), which could not be a problem, but corrections might be necessary for comparison with Nexium (contains 40 mg of esomeprazole, Exptl. section). If the authors measured EtOH to an accuracy of ten micrograms or ten microlitres (of course, they couldn't), then API should have been weighed closer to 20 mg.

- The Hausner ratio and the Carr index are equations but are not identified as equations.

- Just after the equations, 2 kp is written as hardness. What is kp? If that is kilopond, then it is not an SI unit, but if its meaning is different, please define it.

- Table 2 also contains some funny values in the coating layer composition. Let's assume that 10T means ten tablets. The referee has doubts that the author could weigh GMS and P80 with a tenth of mg accuracy. Even the six-digit balances have a serious weighing error in the 1-5 mg range. The pipetting accuracy of water is also doubtful because, in the hundred ml range, the tenth of ml accuracy is impossible and meaningless - it is so, even in the ten ml range. The two decimal digits are out of meaning.

- In Table 3, the first column, the coating layer formatting is unclear. The previous comments on the decimal digits are also valid for Table 3. Please reformat it. Besides this, the referee would put a high bid that weighing, e.g., 1517.2 ml (assuming ten tablets) or even 151.72 ml (one tablet only), is far from the truth.

- Coating weight percentage is also an equation but not marked as an equation.

- In Table 5, the bulk density without mentioning the granulate size is meaningless.

- In general, regarding the graphs, the referee strongly recommend using colored point and curves for clarity.

- Figure 3 is incomplete. The authors later used 180 or 420 min release time, but Fig. 3 does not contain relevant data after 45 min.

- A similar problem in Fig. 4A appears. In Fig. 4A, the release of EMP in I-1 composition seems to increase after ~170 min, but the authors missed extending the experimental time to 420 min. Please resolve the inconsistency.

Figs. B and C must be combined and use colored data points and curves.

- In Fig. 5 caption, the authors missed writing that the pictures are for DR1 or DR2.

- Figure 6 is not in agreement with Fig. 4B/C. Both II-3 and II-6 showed higher differences at a lower resolution than in Fig. 6.

- In the introduction, the authors mentioned that the crucial problem of EMP is the low t½ (1.3 hours). In Table 6, DR1 has a practically identical half-life value compared to the commercial product, which means no improvement. On the other hand, in two-step adsorption, the meaning of t½ needs explanation. The problem is the same for the elimination constant.

- The references do not follow the journal suggestions, as points in the abbreviated journal names are mandatory.

Reference 1 seems incomplete.

Author Response

(The authors gave the same response as above.)

Reviewer 4 Report

Dear authors,

work i very complex and interested. It is great work but it should be more structured in order to be perfect.

I have several suggestions:

Introduction should have somethnig about Male beagle dogs similarity to human

This part should be more aimed> In this study, relates to the manufacture of EMP loaded dual release mini-tablet polycap (DR polycap). Specifically, mini-tablet core formulations containing EMP are prepared and evaluated. The core was coated with ELD-55 as 1 st release mini-tablet or ES100/EL-100 as 2 nd release, respectively. An additional inner coating was required between the core and controlled release coating layer to ensure stability. 1 st release mini-tablet and 2 nd mini-tablet were characterized by in vitro release test and internal and external morphology by using scanning electron microscopy (SEM). The DR polycap is prepared by filling hard capsules with an optimized combination of each mini-tablets (Figure 1). In vivo pharmacokinetic (PK) study was performed for comparing commercial product and prepared DR polycaps with fasting conditions beagle dogs.

Polysorbat 80 surfantant/typo mistake page 6

HPLC method>pH 7.3 phosphate molarity is not reported, it should be stated

selectivity, sensitivity, linearity, accuracy, precision and recovery according - is there supplementary material 

elimination constant (Kel)/---number of time points and what points were selected should be stated

table 2 and 3 are 1ts and snd release minitablet formulation----it is not clearly stated

how formulation of mintablets for DR caps were selected should be more clearly stated

Table 5 was not confirmed that there is no sticking or capping (tabletability value is needed for that) just that flowability is good. Rewrite that text.

core characterisation should be done as well as minitablets

Release profile studies were performed at 25 rpm in Phosphate buffer (pH 6.8), and each formulas profiles are shown in Figure 3.  ------there is no explanation previously for this

Generally work is well written but it should be better organised. In introduction add smth abot human - dog GI similarity. Aim should be shorter. Methodology is not cleary written it should be reorganised.

I understand that complex work is hard to write but this work i great thus just structure it better.

Best of luck

Author Response

(The authors gave the same response as above.)

Round 2

Reviewer 2 Report

There are no further comments.

Reviewer 3 Report

The authors improved their manuscript significantly and exhaustively answered the referee's concerns. The manuscript is suitable for publication.